# Opportunities and barriers in paediatric pulse oximetry for pneumonia in low-resource clinical settings: a qualitative evaluation from Malawi and Bangladesh

Carina King,[1] Nicholas Boyd,[2] Isabeau Walker,[2] Beatiwel Zadutsa,[3] Abdullah H Baqui,[4] Salahuddin Ahmed,[4] Mazharul Islam,[5] Esther Kainja,[3] Bejoy Nambiar,[1] Iain Wilson,[6] Eric D McCollum[7,8]

For numbered affiliations see end of article.

**Correspondence to**
Dr Carina King;
c.king@ucl.ac.uk

## ABSTRACT

**Objective** To gain an understanding of what challenges pulse oximetry for paediatric pneumonia management poses, how it has changed service provision and what would improve this device for use across paediatric clinical settings in low-income countries.

**Design** Focus group discussions (FGDs), with purposive sampling and thematic analysis using a framework approach.

**Setting** Community, front-line outpatient, and hospital outpatient and inpatient settings in Malawi and Bangladesh, which provide paediatric pneumonia care.

**Participants** Healthcare providers (HCPs) from Malawi and Bangladesh who had received training in pulse oximetry and had been using oximeters in routine paediatric care, including community healthcare workers, non-physician clinicians or medical assistants, and hospital-based nurses and doctors.

**Results** We conducted six FGDs, with 23 participants from Bangladesh and 26 from Malawi. We identified five emergent themes: trust, value, user-related experience, sustainability and design. HCPs discussed the confidence gained through the use of oximeters, resulting in improved trust from caregivers and valuing the device, although there were conflicts between the weight given to clinical judgement versus oximeter results. HCPs reported the ease of using oximeters, but identified movement and physically smaller children as measurement challenges. Challenges in sustainability related to battery durability and replacement parts, however many HCPs had used the same device longer than 4 years, demonstrating robustness within these settings. Desirable features included back-up power banks and integrated respiratory rate and thermometer capability.

**Conclusions** Pulse oximetry was generally deemed valuable by HCPs for use as a spot-check device in a range of paediatric low-income clinical settings. Areas highlighted as challenges by HCPs, and therefore opportunities for redesign, included battery charging and durability, probe fit and sensitivity in paediatric populations.

**Trial registration number** NCT02941237.

## Strengths and limitations of this study

► This is the first study to report on end-user perceptions of opportunities, challenges and desirable design features of pulse oximeters used for paediatric pneumonia management in low-resource settings, including community and outpatient providers.

► A key strength was the wide range of healthcare provider perspectives included, from community to referral hospital settings in South Asia and sub-Saharan Africa.

► The study was limited to participant's experience of using specific pulse oximeters and, therefore, may lack generalisability to other paediatric pulse oximeters not used in these settings.

## INTRODUCTION

Several interventions, such as pneumococcal conjugate vaccine (PCV) and standardised guidelines for diagnosis and treatment, have led to reductions in pneumonia morbidity and mortality over the last 20 years.[1 2] However, in spite of these gains, pneumonia remains the leading cause of infectious mortality among children globally, with the vast majority of the burden falling in sub-Saharan Africa and South Asia.[3] To accelerate reductions in pneumonia mortality, further refinement of diagnosis and treatment pathways is needed, including correct referral and access to oxygen treatment.[4]

Pulse oximetry non-invasively measures peripheral arterial oxyhaemoglobin saturation ($SpO_2$). Hypoxaemia (defined as an $SpO_2$ <90%) is included in the WHO guidelines as a pneumonia danger sign,[5] and is associated with increased mortality from pneumonia, as well as other illnesses like malaria.[6–8] Recent evidence from Malawi has also indicated that an $SpO_2$ of 90%–92% is predictive of mortality among children hospitalised with pneumonia.[8]

While some studies have attempted to predict hypoxaemia in children with pneumonia using a combination of clinical signs, there has been mixed success.[9–11] Clinical signs alone fail to identify a proportion of hypoxaemic children based on the current WHO guidelines, which results in a missed opportunity for referral and appropriate treatment.[12 13] In addition, the subjectivity of clinical signs can lead to variation in care—especially among community healthcare workers (CHWs), who often lack ongoing supervision.

Pulse oximeters have been successfully used in low-resource paediatric settings, but are yet to be widely adopted, particularly during outpatient care.[14 15] The Ethiopian Ministry of Health has demonstrated leadership in this area, setting up an initiative in 2016 to ensure oximetry and oxygen therapy are available nationally across the healthcare system.[16] However, Ethiopia is an exception, with implementation of oximetry in other low-income and middle-income countries continuing to be slow. Implementation barriers cited include cost, issues with training and supervision, and the lack of appropriately designed, robust oximeters and universal paediatric probes, although there are several initiatives to develop devices for low-income settings (eg, Lifebox and the Phone Oximeter).[17–19]

In order to better understand current barriers to the use of pulse oximetry by healthcare providers (HCPs) in a range of healthcare settings and explore opportunities that this technology provides, input from end-users is needed.[20] With the ultimate goal of designing a universal paediatric probe for all levels of healthcare services in resource-poor settings, we aimed to gain an understanding of the challenges of pulse oximetry, how its use has changed service provision and how current devices could be improved for these settings. This end-user perspective is currently limited in the literature and is essential to ensure investment in pulse oximetry is sustainable and effective.

## METHODS

We conducted a qualitative study with HCPs from different levels of the healthcare system in one site in Malawi (Mchinji district, Central region) and one in Bangladesh (Sylhet district, Northeast region) from May to July 2016 as part of a wider programme of work aiming to design a universal paediatric oximeter probe.

### Setting

In Malawi, there are three levels of government-provided healthcare: CHWs (locally known as health surveillance assistants), health centres and district hospitals. CHWs conduct weekly or bi-weekly village clinics and home visits providing basic integrated community case management for paediatric infections.[21 22] Health centres are outpatient facilities run by nurses, clinical officers or medical assistants, and district hospitals have inpatient facilities with capacity for oxygen treatment. In Mchinji district,

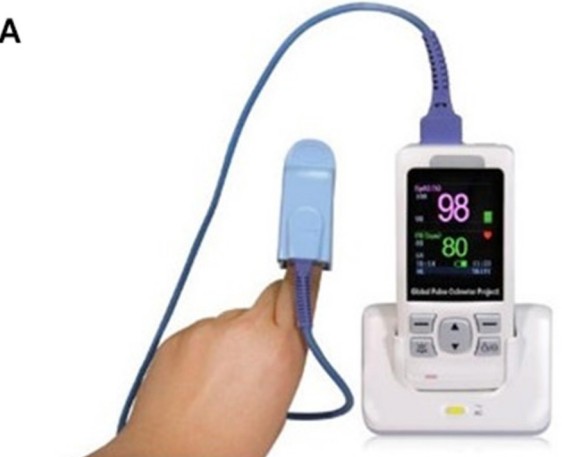

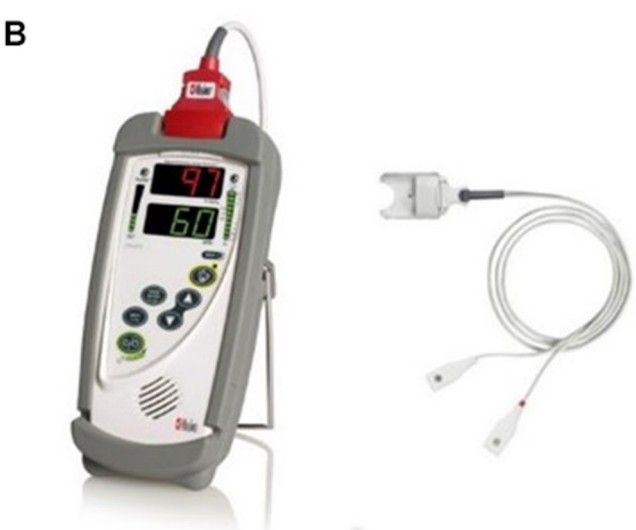

**Figure 1** Pulse oximeters and probes used by healthcare providers in routine clinical care. (A) Lifebox oximeter and adult universal clip probe used in Malawi (accessed on 1 July 2017 from www.lifebox.org). (B) Masimo Rad5 oximeter and LNCS Y-I Multisite wrap probe used in Bangladesh (accessed on 1 July 2017 from www.pacificmedicalsupply.com).

pulse oximetry was successfully introduced into all three healthcare settings in 2012 as part of a PCV research project, using the Acare Technology AH-MX manufactured Lifebox oximeter and universal adult clip probe (figure 1A).[12]

In Bangladesh, the study was conducted at Projahnmo, a research consortium composed of Johns Hopkins University and several local non-governmental organisations in partnership with the Bangladesh Ministry of Family Health and Welfare. Current Projahnmo activities are integrated within three government-operated subdistrict hospitals called Upazila Health Complexes (UHCs), and the referral government hospital in Sylhet city (Osmani Medical College), all of which are staffed by physicians and nurses. The UHCs operate outpatient clinics for children aged under 5 years and provide basic

inpatient paediatric care, including oxygen. The majority of government-provided inpatient care is provided at Osmani Medical College. Female CHWs employed by Projahnmo conduct bimonthly household surveillance, with a subset of CHWs providing weekly surveillance as part of a PCV effectiveness study. Projahnmo CHWs conduct basic clinical assessments and refer ill children for care at the UHCs; they do not administer medicines themselves. Since 2015, a National Institutes of Health-funded study (K01TW009988) trained and supplied all Projahnmo clinical staff in Bangladesh, including CHWs, with pulse oximeters to screen children for hypoxaemia, using the Masimo Rad5 oximeter and the LNCS Y-I Multisite wrap probe (figure 1B).

In Malawi, CHWs individually own the oximeters, and facilities were given a device for each clinic or ward, while in Bangladesh, Projahnmo owns the oximeters and individual HCPs are responsible for routine care and maintenance of the devices. Oximetry was not included in the Malawi paediatric guidelines, and Bangladesh did not have national paediatric pneumonia guidelines at the time of the study.

## Design

We conducted focus group discussions (FGDs). We planned three FGDs in each country, aiming to recruit between 6 and 10 people for each FGD (up to 60 participants in total). This number of groups was agreed on before data collection began, driven by practical considerations given the fact that a limited number of healthcare workers in either setting have experience using pulse oximeters with children. The groups were planned to be CHWs, health centre or UHC staff, and referral hospital staff separately. Conducting separate FGDs for the different types of healthcare workers was to allow context-specific discussions and encourage participants with varying training backgrounds to feel confident about raising challenges relevant to their specific setting.

## Sampling

HCPs were purposively sampled from sites where pulse oximetry had been introduced, and the participants had received some form of training or mentorship in oximetry. Participants were identified by local researchers (BZ in Malawi and SA in Bangladesh) to be a representative sample of HCPs from their setting (eg, small and large health centres, inpatient wards and outpatient departments in the hospital), and contacted directly by phone. All HCPs contacted participated. Participants were reimbursed for their travel costs to the local healthcare facility and provided with refreshments.

## Procedure

FGDs were led by local researchers with experience in conducting qualitative research, with support from a facilitator with knowledge of pulse oximeters. The FGDs were divided into two sections, the first addressing the participants' personal experience with using pulse oximeters.

The topic guide included positive and negative experiences, and possible improvements and challenges (online supplementary web appendix 1). During the second part of the discussion, the participants were presented with different probe designs and given an opportunity to use them for an hour (online supplementary web appendix 2). Following this, the discussion addressed positive and negative aspects of the different designs to encourage critical thinking of possible design solutions to the current limitations of a universal paediatric probe.

The FGDs were audio recorded and then transcribed, along with the facilitators notes. Questions were asked in a mix of English and local dialects depending on understanding and ease of expression (Chichewa, Bangla or Sylheti) and participants were told to answer in their preferred language. Responses were clarified by facilitators if there was an issue with language and understanding between participants. Recordings were transcribed and translated where necessary. Translations for Malawi were done by BZ and EK together until final transcripts were agreed, and by an independent professional service for Bangladesh.

## Analysis

We analysed the data thematically using a framework approach, as an appropriate method for a multidisciplinary team conducting health research.[23] This process involved five steps: familiarisation, identifying a thematic framework, indexing, mapping and interpretation.[24] The transcripts and notes from the FGDs were printed and coded on paper, with the coding matrix created in Microsoft Excel. CK and KF independently familiarised themselves and indexed the data, and the emergent themes were discussed until a consensus was reached on the mapping and interpretation of the data. This interpretation was shared with the local researchers (BZ and EK in Malawi; EDM and MI in Bangladesh) for further discussion until all were in agreement.

Written informed consent was obtained from all FGD participants.

## RESULTS

We conducted six FGDs, with 23 participants from Bangladesh and 26 from Malawi (table 1). We identified five emergent themes: trust, value, user-related experience, sustainability and design.

## Trust

Trust emerged as a theme both in terms of how the HCPs interpret the oximetry results, and how caregivers interact with HCPs and the pulse oximeter. We found that all cadres of HCPs in both sites had an awareness of the fallibility of the oximetry readings, specifically relating to lower $SpO_2$ values. For $SpO_2$ levels which were deemed abnormal, <90% to <95% according to different participants, HCPs stated that they would often recheck the result before making a referral or treatment decision:

**Table 1** Summary of the FGD participants and their work experience

| | Community level | Health centre or Upazila Health Complex | Hospital |
|---|---|---|---|
| **Bangladesh** | | | |
| Total participants | 8 | 7 | 8 |
| Job titles (number) | Community healthcare worker (8) | Physician (4) Medical officer (3) | Senior staff nurse (1) Associate professor (2) ICU staff (1) Anaesthesiologist (1) Assistant registrar (1) Intern medical officer (1) |
| Work experience (mean, range) | 1.7 years (0.6–4) | 2.3 years (1–6) | 14.7 years (0.5–32) |
| **Malawi** | | | |
| Total participants | 9 | 8 | 9 |
| Job titles | Community healthcare worker (8) Vital signs assistant (1) | Medical assistant (7) Medical technician (1) | Clinical officer (3) Nurse midwife (3) Medical assistant (3) |
| Years of work experience (mean) | 10.6 years (5–20) | 8.3 years (3–23) | 8.1 years (4–13) |

FGD, focus group discussion; ICU, intensive care unit.

if we see it is 89% we change the probe or change the finger (Hospital, Bangladesh)

However, questioning the validity of these lower $SpO_2$ results in the context of a child's clinical condition was only discussed by the HCPs who worked in the hospital setting. This difference in the trust placed in the $SpO_2$ results by different types of HCPs suggests that more in-depth clinical training and understanding of the technology may result in different clinical applications:

sometimes the pulse oximeter can give readings which you are not comfortable with according to the presentation of the child…most of the time when it happens like that, we just use our judgement (Hospital, Malawi)

An outcome of using pulse oximeters for pneumonia diagnosis was a change in parental and community understanding and perceptions of care, with HCPs discussing increased trust in their referral and treatment decisions. This worked in two ways, first with the oximeter acting as a direct feedback and education tool:

if the mother is able to read you can show the exact figure and she will accept the treatment of oxygen, [before] it was very difficult to explain the role or the importance of the oxygen machine and some mothers refused (Hospital, Malawi)

Second, in Malawi, HCPs projected that the oximeters had improved clinical care, and therefore outcomes, which led caregivers to be more inclined to accept the referral or treatment being recommended, especially in the case of oxygen:

[previously] in the village they were saying that when a child is put on the oxygen machine it facilitates death, therefore it was making problems, but this time because children are put on oxygen earlier they survive, it's because we knew the saturation (Health Centre, Malawi)

### Value
The theme of value relates to the inherent value of improved decision-making, HCPs perceived self-value (ie, confidence) in their clinical work, and the physical value placed on maintaining a working pulse oximeter. As pneumonia is classified using a range of non-specific and often subjective clinical signs, HCPs discussed the positive addition of this more objective measure:

…by looking at this we can understand how much respiratory distress is in there. Of course this helps us a lot. (Health Centre, Bangladesh)

In both sites, HCPs from front-line settings (CHWs, health centres and UHCs) stated that the pulse oximeters had changed the way they work and given them confidence in making referral decisions. Interestingly however, in the referral hospital setting in Bangladesh where staff training is higher, very little value was placed on the pulse oximeter for improving their clinical performance, with the ability to conduct chest X-rays, lung ultrasound and their clinical judgement valued more highly:

…its sensitivity and specificity is very negligible to be taken as a diagnostic tool. (Hospital, Bangladesh)

In Bangladesh, the CHWs reported pride in using the oximeters. In Malawi, the CHWs placed a physical value on the oximeters and discussed the personal effort, such as paying out of pocket to travel to commercial charging services, they put in to maintaining a working device:

…we have been trying all that is humanly possible to take care of these things, but it only becomes a problem when it comes to the issue of charging. (CHW, Malawi)

This was also reflected at the health centre, where not all facilities have electricity and one or two staff are responsible for assessing children. At the referral hospital however, this was not discussed, with oximeters belonging to the ward, which has a more consistent power supply. Ward-based ownership was discussed as a challenge, suggesting individual ownership could result in improved care and maintenance as having a device in working order would not be dependent on the performance of others:

…some of the clinicians do not take care of them, so when the machine is not working it means the whole department is affected (Hospital, Malawi)

### User-related experience

HCPs at all levels discussed their experiences of using pulse oximeters in children under 5 years, presenting challenges, their solutions and perceptions of usability. The time taken to get a measurement ranged widely, with CHWs in Bangladesh agreeing measurements took less than 1 min but in Malawi that it could take up to 20 min. The factors that increased the time taken to get a measurement were consistently cited as movement and physically smaller children, and in Malawi dirty toes making measurements difficult:

Getting readings from irritable babies is a bit tough and it takes time. (Health Centre, Bangladesh)

…using it on a child up to six months of age, sometimes it has been a problem because these children have got small fingers, so although we use toes sometimes they are also small and the child is afraid so they start crying. So we have got other things we can give a child to play with but it is a little bit of a problem, but at the end we get the results. (CHW, Malawi)

Solutions to these challenging children included asking caregivers to breast feed, giving them a toy to distract them and simply waiting. The term used frequently to describe challenging children was 'fear', with the HCPs stating that children are afraid of having the measurements taken. This fear was associated with the sensors' red light which frightened children, the anticipation of pain or just being an unknown. All of these could result in the child being agitated, crying and uncooperative. Despite these issues in small and agitated infants, the oximeters were considered easy to use:

…it's not complicated, it doesn't need complicated education for a healthcare worker to use, with a good explanation from a colleague or friend you are able to use it. (Hospital, Malawi)

There was also the acknowledgement that time to reading was not as important as getting the correct measurement; for some respondents, the reason some measurements take longer is the desire to get a reliable reading. This included cleaning the child's digits or repositioning the probe:

…taking longer does not mean that one doesn't know the procedure, but sometime it's because you want to give the correct reading. (CHW, Malawi)

A key challenge reported by front-line HCPs in Malawi was about keeping the oximeter charged; this was not considered a significant challenge in Bangladesh. However, here they had issues with ensuring the oximeter remained dry and protected during rains, and being fully waterproof was desirable. Depending on usage, battery life was reported as 1 week–2 months.

### Sustainability

Sustainability was discussed in terms of the device's durability and the need for continued professional development. Generally, the pulse oximeters were thought of as robust and durable, with some of the HCPs having used their device for over 4 years without replacements. However, the battery was highlighted as the least durable part of the device, and there was a perception that when the battery was worn down, the readings became less reliable.

There is a matter with the battery too, if the battery is not enough the reading takes a long time to appear. It sometimes gives false negative readings. (Hospital, Bangladesh)

This related to the HCPs suggestion of having ongoing maintenance support rather than wanting replacement devices. HCPs described the need for ongoing training and support, but also expressed a desire for more in-depth education on how oximetry works which goes beyond the basic training to take a reliable measurement:

A person gets used to what they are doing once they have been oriented. I think sometimes it's also good for you and your team to orient us on how this thing works…the way this thing works, we don't know (Health Centre, Malawi)

In terms of keeping the devices clean and properly stored, an important factor for prolonging shelf-life, we found conflicting opinions between Malawi and Bangladesh. Malawi deemed the probes easy to clean and store securely, although the light colour and materials of the device were thought to show dirt easily. However, in Bangladesh, cleaning was described as burdensome; this likely reflects the different devices and therefore methods

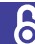

**Table 2** Suggestions of desirable features or improvements given by healthcare providers

| Challenge | Design suggestion |
|---|---|
| Probe fit | Supplied with multiple sizes of probes for different ages |
| | A single cable with multiple probes that can be changed (eg, clipped into the cable) |
| | Softer material for a more comfortable fit |
| Probe placement | Probe made of transparent material so that sensor placement on the nail can be seen |
| Cleaning | Alcohol wipes provided for easier cleaning |
| | A different colour of probe to make it easier to see the dirt, but does not look dirty quickly |
| Power | Solar-powered charger with rechargeable batteries |
| | Back-up power bank |
| | Supplied with a spare battery |
| Agitated children | Toy feature in the device to distract the child |
| | Improve the sensitivity of the device to be quicker |
| | Improve the sensitivity of the device to tolerate movement |
| Integrated spot-check device | Store results in a memory that can be accessed later |
| | Static oxygen saturation result display |
| | '3-in-1' device which includes temperature and respiratory rate measurements as well |
| | Shorter cable length for easier portability |

needed for cleaning, or different perceptions of the importance and frequency of cleaning:

It is hard work to clean it with hexsol and cotton after coming back from the field every day. If we could get something else to clean it with so that we can clean once a week, I don't like cleaning it every day. (CHW, Bangladesh)

### Design

The key challenges mentioned repeatedly across sites and HCP cadres were the battery, sensitivity of the probe in relation to movement or low perfusion, and the probe fit in younger children. Table 2 summarises the design features requested or suggested to improve the pulse oximeters for use in these low-resource settings. Suggestions covered the probe, such as having detachable probes of different sizes, charging and battery life, such as additional power packs and solar charging, and features to help with agitated children.

The oximeters which HCPs used were designed for continuous monitoring; therefore oxygen saturation is not a single static result. This was seen as a negative, with HCPs in both sites wanting the ability to stop at a result and even store measurements (eg, a blood glucose monitor):

Readings would fluctuate if the baby moves. We don't want that. After getting the actual reading it should stay fixed. (CHW, Bangladesh)

In Bangladesh specifically, the CHWs stated a preference for numbers or bars to indicate measurement quality, rather than a dynamic waveform display. This opinion was not reflected in Malawi, which could be a result of using different devices or different training. A

specific challenge presented by CHWs in Malawi was the use of the oximeter in direct sunlight; CHWs often hold clinics outside and they faced the combined challenges of bright sunlight and dust, both of which they reported as challenges in taking measurements:

…it returns the correct results when you are in the shade, but while you are in sunlight it fails to determine good results. (CHW, Malawi)

Positive design features included the portability of devices, the ease of using them and perceived durability, with little direct criticism of the oximeters that the HCPs had been using:

…of the things I like most about using the pulse oximeter, the first one is the portability, because I can use it anywhere. (Hospital, Malawi)

### DISCUSSION

We investigated end-user experiences of using pulse oximeters by a range of different HCPs across clinical settings in Malawi and Bangladesh. The FGDs highlighted similarities in experience, such as challenges in battery durability, the difficulty of small and agitated children and the positive impact of oximeters on clinical practice. However, there were key differences between the providers' experiences in Malawi and Bangladesh and between HCP cadres.

Of note was the difference in perceived ease of cleaning, which was seen as more burdensome in Bangladesh. This is likely associated with the Y-shaped wrap probe design, compared with the more easily cleaned clip design used in Malawi (figure 1). Interestingly though, most critiques

were similar between sites, highlighting some of the major challenges of using pulse oximeter in children— namely movement, low perfusion and small digits. This consistency between our sampled HCPs from each site suggests that these challenges are not device-dependent, and therefore, a specifically designed reusable device for universal paediatric use in low-resource settings is needed.

We identified differences in the sense of value placed on the oximeters by the HCPs, with the higher trained HCPs attributing less value to the results than the HCPs with more limited training. Those with more training valued their clinical judgement more and were more willing to question the accuracy of $SpO_2$ results. This poses interesting lessons for scaling-up implementation and training, as despite the perceptions that obtaining an $SpO_2$ measurement is generally easy, the interpretation of the result is more nuanced. Sustained mentorship and more in-depth training were desired by the HCPs, and this needs to be considered as part of any implementation programme.

As the oximeters were used as spot-check devices rather than continuous monitors, as would generally be found in operating theatres or high-dependency care in high-income settings, many of the suggested design changes related to improving the devices for this process. One example of this was the need for improved battery life and charging, with HCPs highlighting their limited ability to easily access charging points, unlike high-income inpatient settings. Consistently, the desire for quicker, static results and a movement-tolerant probe with improved fit on younger infants was important. Unexpected issues, such as usability in direct sunlight, emphasise the importance of end-user engagement in product development as clinical devices designed for high-income settings would not need to be robust to outdoor use.

The idea of a pulse oximeter being able to improve trust between a caregiver and HCP poses potentially exciting opportunities for improving referral and treatment for paediatric pneumonia. Early diagnosis and treatment as downstream in the health system as possible, ideally to the level of CHWs, are key strategies for improving pneumonia outcomes and therefore reducing morbidity and mortality burden.[25] Therefore, an objective and simple clinical tool with in-built decision support, for example, auditory or visual alarms when the $SpO_2$ is outside of normal range, presents an opportunity for caregiver education and empowerment in the referral decision-making process. Recent data from Malawi supports the potential for oximetry to improve referrals, with HCPs from front-line settings more than twice as likely to correctly refer clinically eligible children with an $SpO_2$ <90% compared with those with an $SpO_2$ >90% during routine outpatient care.[12] Interestingly, this has not necessarily been the case with other more objective diagnostic tools, with examples of rapid diagnostic malaria tests leading to provider–caregiver tensions around treatments.[26 27]

This study was potentially subject to social-desirability bias, with healthcare workers expressing opinions which they thought the facilitators wanted to hear. The purpose of the study was explained to the participants during the consent process and was highlighted as an opportunity for them to contribute to the design of a revised paediatric oximeter and probe. In addition, the groups in some cases were mixed in terms of gender and job titles, possibly influencing participant's confidence in expressing their views and experiences. To mitigate these potential biases, the facilitators encouraged all participants to contribute to the discussions and to be critical throughout. Both positive and negative views were given in both Malawi and Bangladesh, and by different types of HCPs, therefore we do not feel these biases detract from our findings. Finally, we were limited by the number of groups we conducted; additional groups or a different sampling approach may have led to alternative perspectives being included, as the number was not driven by saturation. Therefore, the conclusions we draw need to be interpreted accordingly.

Overall, pulse oximeters were valued by the HCPs we sampled for this study, despite challenges with charging, maintenance and speed of achieving accurate readings in moving or smaller children. This implies that making improvements to currently available oximeters and probes could further facilitate successful implementation of this technology at the community through to the hospital level for routine paediatric care in these two settings. Based on these providers' varied experiences, we recommend that efforts to redesign a pulse oximeter for paediatric spot checks focus on improvements to battery durability, better fit for smaller digits and the speed at which readings are obtained. These were all important challenges which did not necessarily have local solutions presented. More substantive design changes could focus on alternative power and charging systems (eg, solar charging) and '3-in-1' devices which include respiratory rate and temperature measurements.

**Author affiliations**
[1]Institute for Global Health, University College London, London, UK
[2]UCL Institute of Child Health, Great Ormond Street Hospital NHS Foundation Trust, London, UK
[3]Parent and Child Health Initiative, Lilongwe, Malawi
[4]International Center for Maternal and Newborn Health, Johns Hopkins Bloomberg School of Public Health, Baltimore, Maryland, USA
[5]Department of Anthropology, Shahjalal University of Science and Technology, Sylhet, Bangladesh
[6]Lifebox Foundation, London, UK
[7]Eudowood Division of Pediatric Respiratory Diseases, Johns Hopkins School of Medicine, Baltimore, Maryland, USA
[8]Department of International Health, Johns Hopkins Bloomberg School of Public Health, Baltimore, Maryland, USA

**Acknowledgements** We would like to thank all those healthcare providers who took part in the focus group discussions for their time.

**Collaborators** Katie Fernandez, Charles Makwenda, Tambosi Phiri, Tim Colbourn, Mike Bernstein, Nazma Begum, Arun Dutta Roy, Abu Abdullah, Mohammad Hanif.

**Contributors** The qualitative study was designed and topic guides developed by IWi, IWa, CK and EDM, and the field manual written by CK. Oversight of the study was conducted by CK, BN and BZ in Malawi and EDM, AHB and MI in Bangladesh. In Malawi, BZ and EK arranged, conducted, transcribed and translated the focusgroup discussions (FGDs). In Bangladesh, SA and MI arranged and conducted the FGDs.

The data were coded and analysed by CK. The manuscript was written by CK, with considerable input from EDM. IWi, IWa, EDM, BZ, EK, SA, MI, NB, AHB and BN read, commented and approved the manuscript.

**Funding** This study was funded by the Bill & Melinda Gates Foundation (grant number: OPP1133291).

**Competing interests** None declared.

**Patient consent** Obtained.

**Ethics approval** University College London REC (8075/003); Johns Hopkins Medicine Institutional Review Board (IRB00047406); Malawi National Health Sciences Research Committee (16/4/1570); Bangladesh Medical Research Council (BMRC/NREC/2013-2016/1272).

**Provenance and peer review** Not commissioned; externally peer reviewed.

**Data sharing statement** Anonymised transcripts can be shared, following the signing of a data sharing agreement, subject to approval from the relevant national ethics committees. For further information, contact CK (c.king@ucl.ac.uk).

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
