## [Reviewer comments · BMJ Open]

ARTICLE DETAILS

TITLE (PROVISIONAL)	Opportunities and barriers in paediatric pulse oximetry for pneumonia in low-resource clinical settings: a qualitative evaluation from Malawi and Bangladesh
AUTHORS	King, C.; Boyd, Nicholas; Walker, Isabeau; Zadutsa, Beatiwel; Baqui, Abdullah; Ahmed, Salahuddin; Islam, Mazharul; Kainja, Esther; Nambiar, Bejoy; Wilson, Iain; McCollum, Eric

VERSION 1 – REVIEW

REVIEWER	Kevin Baker Malaria Consortium, United Kingdom Karolinska Institute, Sweden I am on the advisory committee for the Lifebox project focused on developing a new pediatric probe.
REVIEW RETURNED	06-Sep-2017

GENERAL COMMENTS	Very well written and interesting paper. Some minor comments and suggestions: Overall comment - would be interesting to see any results included on possible uses for pulse oximetry if these were mentioned - i.e. uses other than screening for severe pneumonia. P2 L48 Think about rephrasing to "Challenges in sustainability related to battery durability and replacement parts were reported, however many HCPs who had used the same device..." P3 L59 This is not a strength or limitation and reads more like a result - perhaps remove or move to the results P5 L128 Add (Figure 1a) after Lifebox P5 L131 Add (Figure 1b) after Masimo P9 L246 Interesting finding on 'Fear' - might be worth expanding on this a little more P11 L290 Panel 1 instead of Box 1 P11 L292 Perhaps expand or mention the key design elements in Box 1 in the text
--

REVIEWER	Miguel Lanaspá Global Health and Tropical Medicine, Instituto de Higiene e Medicina Tropical, NOVA University, Lisbon, Portugal
REVIEW RETURNED	13-Sep-2017

GENERAL COMMENTS	This manuscript constitutes a well-conducted qualitative study on a relevant topic for global health and merits, as far as I am concerned, to be published. The need to improve hypoxemia diagnosis in developing countries based on portable devices is clearly explained and justified, and interviewing users of such devices to detect challenges and preferable designs of improved probes increases the suitability and acceptability of future probes. My only comment concerns the probes that the interviewees were invited to try. Are these probes already used somewhere else? Are they still being developed by companies? An appendix with the characteristics of these probes will allow innovators to detect features where there is still room for improvement. Whether they are commercially available or under development, stating the companies owning the patents will add transparency to this manuscript.
--

REVIEWER	Abigail Enoch University of Oxford, UK
REVIEW RETURNED	14-Sep-2017

GENERAL COMMENTS	This paper is very interesting and explores an important gap in the literature. The title, objectives, and abstract are clear, and for the most part, the writing of the paper is also clear and logical. It is very useful that the topic guide and pulse oximeter figures are included in the paper/appendix. However, there are several areas of concern, particularly regarding the methods and discussion of the limitations: 1. There seem to be some fundamental issues with the composition and number of the FGDs: a) The researchers mention (line 135) that “Conducting separate FGDs for the different types of healthcare workers was to allow context-specific discussions and encourage participants with varying training background to feel confident about raising challenges relevant to their specific setting”. However, according to Table 1, the researchers included HCPs with very different roles in the same FGD, e.g. a senior staff nurse, associate professor, intern medical officer etc. all in one ‘Hospital’ FGD. This is problematic for multiple reasons: First and foremost, as the researchers said, it is important for FGD participants to feel comfortable talking about their beliefs and experiences around the other FGD participants, and this is less likely when (as participants were in some of these FGDs) they are surrounded by others with different levels of training, and who are in different levels of the health system hierarchy. So, it is possible/likely that some of these FGD participants would not have felt as comfortable speaking as if they had been in a group with just their ‘peers’ of HCPs in the same role; the results may therefore have been biased.
---

Also, HCPs such as nurses, anaesthesiologists, registrars, etc. use pulse oximeters for different purposes / in different contexts from each other, because of their defined roles – again it would be useful for them to therefore be in separate discussions with others who use pulse oximeters for similar purposes / in similar contexts. Thirdly, if the researchers had conducted separate FGDs with different HCP groups within the healthcare facility settings, this would have given them the opportunity to compare responses of the different groups.

Along similar lines, it would normally be recommended that separate FGDs are conducted by gender, given that societal/cultural norms and pressures may lead to females not being as comfortable opening up with males present as with only females; this is especially the case when conducting research in low/middle-income countries, such as Malawi and Bangladesh, and yet the researchers included males and females in the same FGDs, again potentially biasing the results.

Ideally, to solve the above issues the researchers would conduct more FGDs, taking the above separations into account. However, if this is not practical then it is important that the researchers at least discuss this problem fully in the limitations section of the discussion.

b) The researchers also only conducted 1 FGD within each healthcare facility context in each country; therefore it is not possible to triangulate the findings of each group, and so there is no way to check whether what was said in one group is representative of e.g. the referral government hospital setting in Bangladesh, vs. an anomaly in the one FGD. It is therefore misleading when, in the findings and discussion, the researchers discuss characteristics/thoughts/experiences of CHWs in Malawi or referral hospital HCPs in Bangladesh etc. – the researchers cannot draw those kinds of conclusions about HCP groups from this data when only 1 FGD was conducted with each of these groups.

There are 2 possible ways to resolve this issue: ideally more FGDs would be conducted; if this is not possible then the discussions of the findings and conclusions should be toned down so that statements are not made differentiating between the groups (for instance the researchers could discuss the findings from Malawi all together, and the ones from Bangladesh all together, and then at the end of the discussion, the researchers could say that from their data, it seems that some of the differences between e.g. CHWs and hospital HCPs are x, y, z, but that they cannot draw more definitive conclusions on differences between the groups because of the small numbers of FGDs).

2. The researchers should provide information on why they decided to conduct that number of FGDs, e.g. was this decided beforehand? If so, why? Did they measure whether they reached saturation? If so, how? And if they did not have any measure or consideration of reaching saturation before they stopped carrying out FGDs then they should mention that as a limitation in the discussion.

3. The researchers should also talk about the recruitment method in the limitations section of the discussion, as recruiting by purposive sampling through identification by local researchers could have led to selection bias.

4. The researchers mention using a framework approach, and identifying a thematic framework (lines 35, 161 and 163) – it is important that they specify which specific theory/model/framework they used (and preferably why).

5. Given that the researchers say they are conducting this research in order to obtain recommendations for creating a paediatric pulse oximeter for use in low-income settings, it seems strange that in the Introduction there is no mention of other efforts there have been to design pulse oximeters specifically for use in low-income settings e.g. Lifebox (especially given one of the researchers is from the Lifebox Foundation), ones to be used with smartphones, etc., and perhaps why these were felt to not serve the population well enough, hence the need for this research.

6. The researchers should be careful about making too many generalizations about “Malawi” and “Bangladesh” given that all of the FGDs were conducted in one geographic area in each country. The results are interesting because they are examples of thoughts/experiences etc. from low-income settings, but they cannot be said to be representative of those two countries as a whole, and this should be stated somewhere.

7. On line 125, researchers state “Currently pulse oximetry is not part of standard care in the community or health centre setting in either Malawi or Bangladesh.” It would be helpful if the researchers could clarify this statement (e.g. not standard care based on guidelines? Or numbers of HCPs using them? Should specify and provide references), as at the moment, the statement is confusing since directly after it, the researchers write that pulse oximeters were successfully introduced / supplied with training to both settings.

8. Line 155: the researchers write that “The participants were told to answer in English or their native tongue”. a) In what language were the questions asked? This should be stated. b) Surely if participants could answer in a range of languages then other participants in the FGD may not have been able to understand their answer? In which case this may have limited the discussions? Or were all answers then repeated back to everyone else, translated in to English (or another language)? Currently this is unclear for the reader.

9. Line 221: The researchers state that the CHWs in Malawi individually own the pulse oximeters --- it would have been useful to have included this information in the description of the different types of HCPs in the Setting section of the Methods.

10. The issue of HCPs in Malawi needing to travel to charge their pulse oximeters, and that this costs them money is discussed in both the Value section (line 221) and in the User-related experience section (line 259). This repetition is a bit cumbersome; it would be better if this was just mentioned in one section, or if the points were differentiated more.

11. Line 228: The researchers write “... suggesting individual ownership could result in improved care and maintenance”. This is a confusing statement, given that the researchers have just discussed how HCPs who own their own pulse oximeters spoke about difficulties in care and maintenance of these due to e.g. costs of charging.

	12. Line 280: the description of HCPs trying to keep the devices clean and finding this burdensome seems more suited to the User-related experience section. It would be helpful if the researchers could perhaps clarify how this is more related to Sustainability. 13. Line 345: the researchers write "...presents an opportunity for caregiver education and empowerment in the referral decision-making process. Recent data from Malawi supports this notion. ..." However, the results stated after this statement do not seem to be about caregiver education and empowerment but rather about what HCPs do with pulse oximeter results. It would be helpful if the researchers could clarify how these results support the previous statement. 14. The second point in the Strengths and Limitations section is not really a strength or a limitation, just a description of some of the results. 15. Box/Panel 1, Line 463: It would be helpful if the researchers could clarify in the title of this Box/Panel whether these suggestions are those made directly by the HCPs or if these are suggestions the researchers are making based on what the HCPs discussed. More minor considerations: 1. In line 43 and 177, you miss a semi colon after "sustainability"; as it reads now, there are only 4 emergent themes.
--	--

VERSION 1 – AUTHOR RESPONSE

Reviewer: 1

1.1 Overall comment - would be interesting to see any results included on possible uses for pulse oximetry if these were mentioned - i.e. uses other than screening for severe pneumonia.

This is a good point – part of the informed consent process we presented the project in the context of oximetry for paediatric pneumonia assessment, and HCP training and experience of oximetry was mostly related to pneumonia. None of the HCPs discussed oximetry outside of this context, probably influenced by our objectives, but also a lack of wider training on the uses of oximetry or causes of hypoxemia. This would be something interesting to pursue though.

1.2 P2 L48 Think about rephrasing to "Challenges in sustainability related to battery durability and replacement parts were reported, however many HCPs who had used the same device..."

Thank you for the suggestion - this has been amended

1.3 P3 L59 This is not a strength or limitation and reads more like a result - perhaps remove or move to the results

This has been removed

1.4 P5 L128 Add (Figure 1a) after Lifebox; P5 L131 Add (Figure 1b) after Masimo

Thank you for picking these up, now included

1.5 P9 L246 Interesting finding on 'Fear' - might be worth expanding on this a little more

We agree, fear was commonly raised as causing children to be distressed, and therefore difficult to get a measurement from. We have expanded on this, and it now reads: "The term used frequently to describe challenging children was 'fear', with the HCPs stating that children are afraid of having the measurements taken. This fear was associated with the sensors' red light which frightened children and the fear of the measurement causing pain, or just being unknown. All of these could result in the child being agitated, crying and uncooperative."

1.6 P11 L290 Panel 1 instead of Box 1

Thank you for picking this up, we have amended it.

1.7 P11 L292 Perhaps expand or mention the key design elements in Box 1 in the text

Thank you for the suggestion, we have included the following statement: "Suggestions covered the probe, such as having detachable probes of different sizes, charging and battery life, such as additional power packs and solar charging, and features to help with agitated children."

VERSION 2 – REVIEW

REVIEWER	Kevin Baker Malaria Consortium, United Kingdom Karolinska Institute, Sweden I am on the advisory committee for the development of the Lifebox probe but it is not an active group so don't see any conflict.
REVIEW RETURNED	13-Oct-2017

GENERAL COMMENTS	Thank you for your responses and amendments - I think it reads well now and no additions from me.
---

REVIEWER	Miguel Lanaspá Global Health and Tropical Medicine, Instituto de Higiene e Medicina Tropical, NOVA University, Lisbon, Portugal
REVIEW RETURNED	17-Oct-2017

GENERAL COMMENTS	I have no further comments. Congratulations for the paper.
--

REVIEWER	Abigail J Enoch University of Oxford, UK
REVIEW RETURNED	30-Oct-2017

GENERAL COMMENTS	Thank you for your responses to the initial comments. The paper is much improved after the revisions. I only have two minor comments at this point: 1. Referring back to my original comment #5, I appreciate that it is not within the scope of your study to carry out a literature review of the other efforts that there have been to develop pulse oximeters specifically for use in low-resource settings, but I think it is important to have a sentence or two in your introduction or discussion where you mention that other efforts have been made already; otherwise those who are not familiar with the literature may think that this is the first time that anyone has thought to develop a pulse oximeter for low-resource settings, which would be misleading. I see that it was a Lifebox pulse oximeter that was introduced in Malawi, so when you mention this, you could say that this pulse oximeter was developed for low-resource settings. Was the Masimo Rad5 oximeter that was used in Bangladesh designed for low-resource settings? It would be useful to mention whether it was or was not. 2. Referring back to my original comment #13, the wording of the study described in line 354 is still confusing. The researchers write that the study "supports the potential for oximetry to improve referral decision-making". However, the subsequent wording suggests that the study shows that referrals are more accurate for children with low SpO2 values than for children with high SpO2 values, rather than showing that referrals are more accurate when pulse oximeters are used than when they are not used? It would be helpful if this could be clarified.
--

VERSION 2 – AUTHOR RESPONSE

Thank you for the further review of our manuscript, we have no additional revisions or responses to reviewers 1 and 2, and thank all the reviewers for their kind comments.

Specific responses and revisions to reviewer 3's comments are below:

1. Referring back to my original comment #5, I appreciate that it is not within the scope of your study to carry out a literature review of the other efforts that there have been to develop pulse oximeters specifically for use in low-resource settings, but I think it is important to have a sentence or two in your introduction or discussion where you mention that other efforts have been made already; otherwise those who are not familiar with the literature may think that this is the first time that anyone has thought to develop a pulse oximeter for low-resource settings, which would be misleading. I see that it was a Lifebox pulse oximeter that was introduced in Malawi, so when you mention this, you could say that this pulse oximeter was developed for low-resource settings. Was the Masimo Rad5 oximeter that was used in Bangladesh designed for low-resource settings? It would be useful to mention whether it was or was not.

RESPONSE: Thank you for the clarification on this point, we have now included a sentence in the introduction as per your suggestion (lines 90-93). The Masimo device used in Bangladesh was not specifically designed for low-income settings, and the Lifebox oximeter used was designed for surgical use in low-resource settings, rather than specifically addressing the need for a spot-check device for paediatric pneumonia.

2. Referring back to my original comment #13, the wording of the study described in line 354 is still confusing. The researchers write that the study "supports the potential for oximetry to improve referral decision-making". However, the subsequent wording suggests that the study shows that referrals are more accurate for children with low SpO2 values than for children with high SpO2 values, rather than showing that referrals are more accurate when pulse oximeters are used than when they are not used? It would be helpful if this could be clarified.

RESPONSE: Thank you for raising this clarification, you are correct, we mean that with pulse oximetry the referrals are more accurate as those with hypoxemia were more likely to be referred compared to children with normal oxygen saturation, amongst those that were clinically eligible for referral. We have re-worded this sentence (lines 355-358).

VERSION 3 – REVIEW

REVIEWER	Abigail J Enoch University of Oxford, UK
REVIEW RETURNED	05-Dec-2017
GENERAL COMMENTS	Thank you for your responses and revisions. I have no further comments.